# Primary motivations for and experiences with paediatric minimally invasive tissue sampling (MITS) participation in Malawi: a qualitative study

Sarah Lawrence,[1] Dave Namusanya,[2,3] Sumaya B Mohamed,[4] Andrew Hamuza,[5] Cornelius Huwa,[6] Dennis Chasweka,[6,7] Maureen Kelley,[7,8] Sassy Molyneux,[7,8] Wieger Voskuijl,[7,9] Donna Denno ,[7,10] Nicola Desmond [5,11]

SL and DN are joint first authors.

**Correspondence to**
Dr Nicola Desmond;
nicola.desmond@lstmed.ac.uk

## ABSTRACT

**Objective** To understand family member consent decision-making influences and experiences in Malawi in order to inform future minimally invasive tissue sampling (MITS) studies.

**Design** Qualitative study.

**Setting** Queen Elizabeth Central Hospital (QECH) in Blantyre, Malawi, which serves as the central referral hospital for southern Malawi and where MITS participants were recruited from.

**Participants** Families of paediatric MITS participants.

**Methods** We conducted in-depth interviews with 16 families 6 weeks after the death of paediatric MITS participants. Data were analysed using a combination of thematic content and theoretical framework approaches to explain the findings.

**Results** Improved cause of death (CoD) ascertainment was the principal motivator for participation to protect remaining or future children. Community burial norms, religious doctrine and relationships with healthcare workers (HCWs) were not reported influencers among family members who consented to the procedure. Primary consenters varied, with single mothers more likely to consent independently or with only female family members present. Clear understanding of MITS procedures appeared limited 6 weeks postprocedure, but research was described as voluntary and preconsent information satisfactory for decision-making. Most families intended to share about MITS only with those involved in the consent process, for fear of rumours or judgement by extended family members and the wider community.

**Conclusion** Among those who consented to MITS, decision-making was informed by individual and household experiences and beliefs, but not by religious affiliation or experiences with HCWs. While understanding of the MITS procedure was limited at the time of interview, families found informed consent information sufficient for decision-making. Future MITS studies should continue to explore information presentation best practices to facilitate informed consent during the immediate grieving period.

## INTRODUCTION

Improved cause of death (CoD) determination is essential to tailor intervention

## STRENGTHS AND LIMITATIONS OF THIS STUDY

⇒ This study conducted in-depth interviews with families following minimally invasive tissue sampling (MITS) participation to improve understanding of MITS decision-making factors, experiences with the consent process, and plans for MITS results utilisation.

⇒ The MITS in Malawi study was paediatric and hospital-based, therefore findings may not be generalisable to adult or community-based MITS studies or to settings outside of southern Malawi.

⇒ We lack in-depth data from families who declined to consent to MITS, hence our detailed findings only reflect the views shared by family members who consented to the study.

development and implementation to reduce preventable child deaths. Complete diagnostic autopsy (CDA) is the gold standard for determining CoD but is often challenging or unfeasible due to resource constraints, limited acceptability linked to cultural and social norms, and concerns about body disfigurement. In response to these barriers, minimally invasive tissue sampling (MITS) has been validated and is increasingly being used as an alternative to CDA to establish CoD in various settings, including among child deaths.[1 2] MITS uses small needles to sample organs and body fluids for histopathological and microbiologic investigations to inform CoD determinations.[3]

Due to the sensitivity of the procedure, the immediate personal tragedy faced by parents/caregivers from whom consent is sought, and the need to evaluate context-specific acceptability, pre-implementation formative acceptability research has been widely implemented.[4–11] MITS has generally been found to be more acceptable than CDA across diverse settings, reducing some of the

cultural, social and structural restrictions that constrain CDA implementation.[4–6 12 13] In particular, body disfigurement—a commonly cited reason for declining CDA—is mitigated.[4–6 13] Formative research to understand *hypothetical* paediatric MITS acceptability in Malawi, found that acceptability would likely be strongly influenced by social relationships between parents, extended family, hospital workers and research staff, but that religious affiliation would be less influential than individual and household beliefs and practices.[14] Participants believed MITS participation would be facilitated by desire for improved CoD information but hindered by fears of organ harvesting and disruption to transportation and burial plans, corroborating evidence from similar studies in other settings.[10 15] Body disfigurement was still raised as a concern in the Malawi formative assessment.[14]

While evidence on MITS acceptability derived from pre-implementation assessments has increased in recent years, postparticipation assessments have been limited.[8] With increasing MITS utilisation globally, it is important to assess how families perceive the procedures after participation to optimise consent and support processes for grieving families.

## METHODS
### Study setting
The MITS in Malawi (MiM) study was conducted at Queen Elizabeth Central Hospital (QECH) which serves as the central referral hospital for southern Malawi. Services are largely provided free of charge. Eligibility included children (7 days to 4 years) who died during hospitalisation with acute illness and/or malnutrition and with a primary caregiver residing within a 50 km radius of QECH. This catchment area includes ethnically diverse, predominantly Christian, matrilocal urban and rural communities.

MiM started as a substudy of the Childhood Acute Illness and Nutrition network, which aims to identify mortality risk factors among children hospitalised with acute illness or undernutrition across nine sites in six countries,[16] but was expanded to patients enrolled in two other undernutrition studies conducted by the same research group and to the general paediatric wards. Caregivers approached for enrolment were offered transportation and a coffin after information about the study was provided, irrespective of participation decision.

### Study design and sampling
In-depth interviews were conducted with family members who consented to the MITS procedure to understand experiences with the consent process, decision-making factors and their plans for using and sharing MITS results. During the consent for MITS, written consent was obtained from the family for interviews 6 weeks later at QECH or at their home, whichever they preferred. Transport for participants to and from the interview was provided by the study. Interview participants were determined by the family and family members interviewed

together as is natural in the cultural context (see table 1). Those who did not consent to MITS were asked their reasons for withholding consent.

A discussion guide (online supplemental appendix 1) was developed in English, based on formative qualitative research exploring MITS acceptability in Malawi.[14] It was translated to Chichewa (the local language). Sociodemographic data were available from linked case report forms for participants coenrolled in other studies and additional sociodemographic data were collected during the interview. Interviews focused on experiences within the hospital prior to death, interactions with the MITS study team, consent procedures, motivations for participation and intended use of MITS results.

### Data collection and analysis
Interviews were conducted between September 2018 and December 2019. All participants preferred to receive MITS results and be interviewed at QECH. Participants first met with the study team to receive CoD results. They then provided verbal consent to be interviewed. Interviews were conducted by a male, masters-level social scientist (DN, shadowed by AH) who had not previously met the participants and was introduced by the clinical MITS study team member who provided CoD results and answered any questions about the findings. Interviews were conducted in Chichewa largely using a narrative approach led by participants while addressing discussion guide topics. Interviews were recorded, transcribed and translated verbatim. Postinterview reflections (by DN) were noted to capture key topics and initial thoughts.

Transcripts were analysed using a codebook that was developed iteratively through reading postinterview reflection reports and transcripts, code development and refinement, preliminary code application to transcripts, discussion of code application and revision to the codebook. Transcripts were imported into NVivo V.11 Plus (QSR International Pty Ltd). Each transcript was coded independently by one team member (DN, SL, SBM). Transcripts were exchanged and coded transcripts were reviewed by a second coder. Coding disagreements were resolved through group discussion.

Interviews were analysed using thematic content and framework analysis to produce a description of key themes across all interviews.[17 18] Coded data were grouped into themes and then explored by sociodemographic features, such as primary caregiver, past experiences in research and religion to identify if key characteristics influenced MITS decisions, participation experiences and reflections following CoD results provision. Individual case study descriptions were also produced to explore situated experiences in-depth, drawing on a phenomenological approach that recognises the unique experiences of individuals, especially where there is little prior knowledge available about the topic.[19]

| Table 1 | Characteristics of participants (n=15) |
|---|---|
| **Characteristic** | **Median (IQR) or n (%)** |
| Relationship to MITS participant | |
| Mother and father | 5 (33.3) |
| Mother | 4 (26.7) |
| Mother, grandmother, & aunt/uncle | 3 (20.0) |
| Father & uncle | 1 (6.7) |
| Uncle | 1 (6.7) |
| Stepmother and grandparents | 1 (6.7) |
| Prior child death in family | |
| Yes | 1 (12.5) |
| No | 4 (26.7) |
| N/A—first child | 4 (26.7) |
| Data unavailable* | 6 (40.0) |
| Primary caregiver's religion | |
| Pentecostal | 5 (33.3) |
| Catholic | 3 (20.0) |
| Protestant | 3 (20.0) |
| Apostolic | 2 (13.3) |
| Islam | 1 (6.7) |
| Data unavailable* | 1 (6.7) |
| Prior research participation | |
| CHAIN or other nutrition study | 8 (53.3) |
| Other research prior to admission | 1 (6.7) |
| Data unavailable* | 6 (40.0) |
| Reported informed of child's diagnosis during life | |
| Yes | 6 (40.0) |
| No | 7 (46.7) |
| Data unavailable* | 2 (13.3) |
| Characteristics of MITS participant | |
| Female | 4 (26.7) |
| Age (months) | 7.6 (3.0–16.3) |
| Living twin | 2 (12.5) |
| Primary caregiver | |
| Mother | 6 (40.0) |
| Mother and father | 8 (53.3) |
| Grandparents | 1 (6.7) |
| Number of living biological siblings | |
| 3+ | 5 (33.3) |
| 2 | 2 (13.3) |
| 1 | 1 (6.7) |
| 0 | 6 (40.0) |
| Data unavailable* | 1 (6.7) |
| Inpatient stay duration (days) | 4.0 (2.0–8.0) |

*Not discussed in interview with family.
CHAIN, Childhood Acute Illness and Nutrition; MITS, minimally invasive tissue sampling; N/A, not applicable.

**Patient and public involvement**

Patients or the public were not involved in the design, or conduct, or reporting, or dissemination plans of our research.

**RESULTS**

Of the 58 families approached for participation in MiM, 29 (50%) consented (figure 1). Families most often cited lack of benefits as the primary refusal reason—if they were already satisfied with the CoD information provided—followed by not wanting to delay burial, and religious concerns including beliefs that death is God's will and therefore needs no further investigation or that postmortem procedures are explicitly prohibited by their specific place of worship. Two families were disappointed in the care their child received during hospitalisation and felt more should have been done while the child was alive, not research after death. Two other families were unable to reach the family member with the authority to consent (as defined by the family—father and uncle for one case and biological mother for another) and another family felt their child was too young (2 weeks old) to have postmortem procedures conducted.

Of the 29 families who consented to the MITS procedures, 16 participated in interviews. Nine families were not approached for interview due to social scientist unavailability (cases 21–29) and three could not be reached for interview and results provision. Sixteen of sixteen families approached for interviews consented. One interview was omitted due to poor audio quality. Interviews included one to three family members as preferred by participants, most often the mother and father together if both were available (table 1). Interviews lasted between 31 and 91 min.

Participant characteristics of each are summarised in online supplemental table 1, including reasons for participating in MITS and consent dynamics. Sociodemographic characteristics, such as primary caregiver and religion, were unrelated to factors described as influencing decision-making, among those who participated.

For those who consented, MITS participation was largely reported to be influenced by desire for improved CoD knowledge, individual beliefs about burial traditions and religious practices, and in some instances, transportation and coffin provision. Participants reported that interactions with healthcare workers (HCWs) or researchers, either positive or negative, did not influence their decision-making. Primary consenters were identified by families and varied depending on each particular situation. Clear understanding of the MITS procedure was often limited at the time of interview. Most participants did not intend to share MITS results outside of those involved in the consent process. The strength of themes varied across interviews (figure 2).

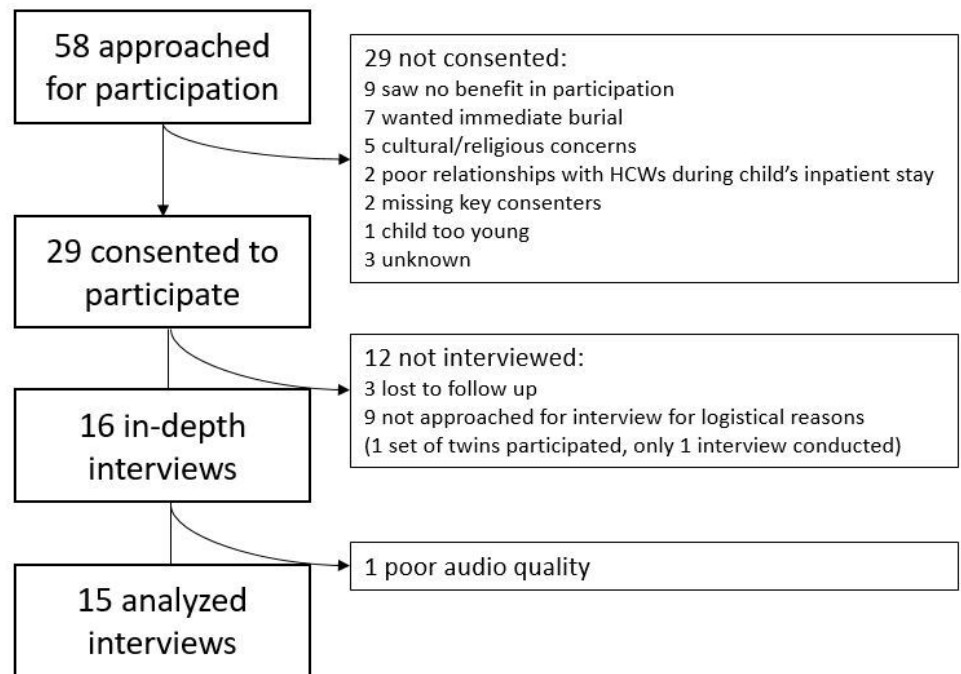

**Figure 1** Consent to minimally invasive tissue sampling and interviews. HCWs, healthcare workers.

## Reasons for MITS research participation
### Knowledge-seeking

Desire for improved CoD information was the primary driver of MITS participation. Caregivers saw MITS as an opportunity to learn about the cause of illness and subsequent death, especially when a clear, well-understood diagnosis was not reported as provided during the child's life. Most families provided with diagnoses during the child's life believed the information was insufficient, due to limited information provided by HCWs or their own

limited understanding. They often believed this prior information was not representative of the 'true' CoD.

> … if we refused [MITS], we would not have known the cause of the death because we were just thinking that it's just mere diarrhea. We didn't really know the disease, so I wanted to know the cause of death. (Mother, Family 2)

For most families, CoD information-seeking was driven by a desire to protect remaining or future children

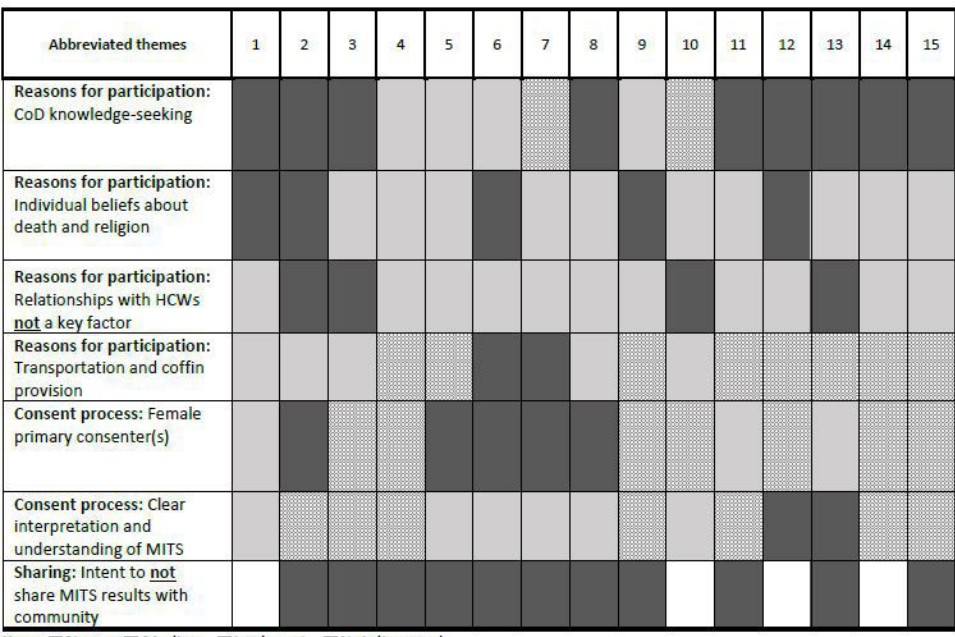

**Figure 2** Strength of themes by family. CoD, cause of death; HCWs, healthcare workers.

through identification of hereditary illnesses or insights that might guide child-rearing practices. Two families reported motivation to protect the life of the living twin, both of whom also had two or more other living siblings.

> We were grappling with questions as parents and we were wondering that with this sort of death, then how best can we take care of the surviving twin child. So we said, let us consent and then they can find out the actual cause of death and if it will help others then also it will help us as parents. (Father, Family 1)

Five participants also identified MITS as beneficial to the wider community through its potential to inform the development of future treatments so others will not have to suffer the loss of a child.

> We agreed to do postmortem so that the findings could help someone in future. We don't want what happened to us to happen to other people. (Father, Family 13)

### Individual beliefs about burial and religion
Participants shared that their decision to participate was strongly influenced by individual beliefs about death and religion rather than by broader community burial norms or religious doctrine.

### Burial
Participants reported they were not concerned with potential burial delays or morticians cleaning the body, instead of following traditional burial practices whereby the family cleans the body prior to burial. Practices regarding burial traditions were seen as guidelines, more than rules.

> We are used to washing the body before burial so when we arrived at the village and the people asked about washing the body, we told them that the body had already been washed and they said that it is not a problem. (Mother, Family 1)

Some families whose child died in the late afternoon elected to wait for the MITS procedure to be conducted the next morning, even though burials are usually conducted on the same day. These participants were particularly motivated to learn CoD information that they hoped MITS would provide and felt that travel by night would be logistically difficult. They did not view waiting for MITS as a significant breach of tradition. Although, ensuring prompt burial was the primary reason stated for refusing participation in MITS for some families (figure 1).

### Religion
Interviewees viewed MITS participation benefits as outweighing community expectations and norms defined by particular religious affiliation. For example, one family with membership in an Apostolic church whose members usually oppose biomedical healthcare chose to both seek inpatient and outpatient medical treatment when their child was severely burned, but to also participate in MITS to learn more about the child's death.

> They [church leaders] say that do not go to the hospital but it's up to you the patient to decide based on how you are feeling … you just discipline yourself [after going to the hospital]. (Mother, Family 10)

Religious affiliations were not known for all of the five families who declined MITS and cited religious or cultural reasons as their primary concern, although one, for example, noted their Presbyterian church does not allow procedures on the deceased. Another noted they do not believe further investigation is needed after death because death is the will of God.

### Relationships with HCWs
Communication with HCWs and research staff, regardless of whether deemed to be satisfactory, did not emerge as a primary factor in consent decision-making for participants. They had variable experiences with HCWs during the QECH inpatient stay and at prior health facilities. Most participants noted some rudeness, limited communication, or lack of follow through by HCWs at some point in their child's care but were ultimately motivated by improved CoD understanding. Participants whose child was enrolled in research studies during life mentioned that researchers exhibited more compassion and attentiveness toward them than clinical staff on the wards.

> When I came to the hospital and met with doctors from CHAIN, I could see that they would talk to me with respect; yet when I met with the others, the doctors from the wards, they just spoke to me rudely. They just spoke as if they are tired with their job. (Father, Family 1)

Participants acknowledged they did not usually ask questions of HCWs on the wards, rather they merely observed, because HCWs have authority over information regarding their child's treatment.

> They did not tell us [about the child's treatment], we were just watching being that they are the doctors. (Grandmother, Family 8)

A few of the participants shared they were counselled about their child's deteriorating health and imminent death, although others voiced concerns that HCWs had not responded to their child's deterioration.

While families had mixed experiences with HCWs prior to the child's death, none identified this as a primary factor in deciding to consent to MITS. However, two families declined MITS due to frustration with the care provided during the child's life.

> Nobody came, they were just busy playing with their phones, but they saw that I was stressed up with the child, they were just looking at him, maybe they knew that the child was dying …. (Mother, Family 10)

### Transportation and coffin provision

HCWs involved in care during the inpatient stay introduced the family to MiM research team. A few families noted they were advised by HCWs that the MiM study presented an opportunity for transportation and a coffin, and this strongly influenced their consent decision-making.

> What made me to accept was that I wanted to know the cause of the death because he was having diarrhea but also, I wanted to be helped with transport to get home. (Mother, Family 6)

When the research team became aware of this practice by some HCWs partway through the study, a hospital staff re-training was implemented to reinforce that transport and coffins would be provided regardless of consent and offered to the families by the research team after approach for consent was complete. The finding of this influencer was no longer observed in interviews conducted subsequent to the re-training (post Family 11).

### MITS consent dynamics
#### Primary consenters

Household norms and familial support in the hospital informed who was involved in the MITS consent process and who retained the ultimate authority to consent or not. Most primary caregivers consulted other family members during the decision-making, often including the child's grandparents, uncles and aunts—especially when they lived nearby or were involved in the child's care during life. When multiple family members were involved, the primary decision-maker for MITS often was the male head of household or the child's maternal uncle.

> I could not give consent on this on my own … we [researcher and grandmother] left together with my aunt and sat beside the mortuary, we [grandparents] linked up. After linking up, it was found that the dad [child's grandfather] agreed and said he had given consent on this. (Grandmother, Family 14)

Some parents did not consult with extended family members, as they believed they held the sole responsibility of their child, including after death. Place of residence (rural vs peri-urban) did not influence the primary consenter choice within the family.

> We did not go to consult because we are the parents, the responsibility towards the child rests with us; that is why we just accepted that they should go ahead because we also wanted to know what the problem was. (Father, Family 1)

Mothers were more likely to consent alone if the father was uninvolved in the child's care generally or if there were marital difficulties.

> I did everything on my own. There was nobody to discuss with. I was like a father and a mother, so I was supposed to make decisions on everything. (Mother, Family 2)

Single mothers were also more likely to only engage women in the consent process, often their mothers and sisters who were supporting them during the child's hospital stay or first to arrive. One woman who had a strained relationship with the child's father consented with the child's maternal grandmother present, who had been with her during the child's hospitalisation, and then later informed the father of the decision, but she did not feel he needed to be involved in the process.

### Interpretation and understanding

Interviewees were often unable to describe the MITS procedure. The only component one participant recalled (erroneously) was the removal of the brain, noting that this was necessary to determine the CoD. She reported satisfaction with the information provided during the consent process but could not recall further details. Another participant described the procedure as an 'X-ray'. Two participants highlighted MITS would test more than one area of the body—one noted three to four areas would be tested, while another that 'each and every part of the body' would be tested. Only two participants explicitly noted that MITS involves needle sampling and no incisions.

> They explained to me saying: if possible, there is need to examine the bodies of the children, we will not tear the bodies in any way, but we just use some needles. So that, once we find out about something in the future, since we still have a generation, maybe this can be something to say: we should get prepared now, before something happens. Therefore, according to the way they explained, I felt they gave in some good points. (Uncle, Family 12)

> The researchers said their research is not about cutting the body anywhere. They said they will use an injection to pull out fluid from the nerves or from the spinal cord. They said they will get these samples from 3 or 4 areas. When they get these samples, they will tell us the results after one month. I told them, there is no problem. They said after taking the samples, we will ask you to go and confirm that we have not cut anywhere. (Father, Family 13)

Across all the participants who described the consent procedures, they understood MITS aimed to provide additional information about the child's CoD and their participation was not mandatory.

> So the hospital people came saying that if the death happening like this [you can participate in MITS] but it's not a compulsory thing. We take the body to the mortuary for MITS to find out the real cause of the death. (Mother, Family 8)

## Sharing MITS results

Participants reported being satisfied with the CoD results received from the MITS research team and were able to explain the findings and recommendations provided. While many participants who consented to MITS identified better understanding of their child's CoD as protective for their households and communities, very few of them planned to share information about their participation or results with extended family, neighbours or other community members. Individuals involved in the consent process were frequently the only individuals with whom parents and caregivers planned to share the results. Participants shared this decision was influenced by fear of what people might say about the procedures and the family themselves.

> Somebody who may not be close, once you tell them, instead of them to understand, they begin asking you questions (Grandmother, Family 14). And they can even tell you that you could be the one responsible for my child's condition, nowadays. (Stepmother, Family 14)

One participant noted that by telling even one relative, the family risked having their decision shared across the community when that relative tells others. Due to concerns that this potential for gossip would cause issues for the family, he decided not to share about MITS.

A couple of participants shared they would not have told their family members about MITS had they not already been present at the hospital. Some participants reported telling extended family they had enrolled in a study that was providing transport but did not elaborate further on research activities to avoid questions or judgement. Relatives and community members did not push for additional details because transportation for deceased is often a challenge and this answer avoided what participants considered unwelcome questions.

Only four families informed uncles, traditionally regarded as heads of the extended family, that MITS had been conducted.

> I will tell members of my extended family since they are my nephews and nieces … so they should know what caused the child's death. (Father, Family 10)

The four families also reported they intend to share information learnt about the CoD with these uncles. Of the four, one participant was primarily motivated to share results to prevent blame within the family.

> People should be able to understand to say: it happened this way, the main issue was that this person died of this problem, so that there should not be finger pointing towards one another. (Uncle, Family 12)

## DISCUSSION

Our findings demonstrate that individual experiences and familial relationships are central to MITS consent and participation processes. Sociodemographic characteristics did not clearly influence decision-making themes, consent processes or intention to share results. One notable exception was that single mothers were more likely to consent to MITS independently or with only female family members present, possibly explained by the matrilocal setting.[20] In cases where the father was at least somewhat involved, they were engaged in the consent process and often had the principal authority to consent, corroborating hypothetical findings from formative MITS acceptability research in Mozambique, South Africa, Bangladesh, India and Pakistan.[6 8 9 21 22] Interestingly, relationships with HCWs or research staff (neither positive nor negative) or dissatisfaction with the healthcare provided, did not seem to strongly influence decision-making among MiM participants, even though many did report negative interactions. This is in contrast to qualitative work pre-MiM study implementation from our group and other formative studies that suggested trust in the healthcare system and relationships with HCWs and research staff would be important drivers of consent.[14 21] However, the data from our in-depth interviews are only among those who *consented to MITS* and reflect families' experiences 6 weeks after the death and MITS procedure.[4 5 14] Nevertheless, for these families, it was apparent that desire for improved CoD knowledge outweighed any negative experiences during the hospital stay and participants seemed to accept limited communication with HCWs as the norm. Participants who had received a clear diagnosis of the child's illness prior to death continued seeking additional information after death in a continuous search for meaning, reflecting similar qualitative findings in Tanzania where families continue negotiation in defining illness after death.[23]

While participants often identified MITS results as beneficial to surviving children and potentially protective to the broader community, they frequently did not intend to share their participation experiences or results outside of close family members involved in the consent process. Families were fearful of being asked unwanted questions, rumours about their family, and judgement about their participation. This reflects similar findings to formative MITS research in Pakistan that found women fear unexpected CoD results or that new, stigmatised diagnoses would result in blame toward them and the possibility of abandonment by their husbands.[9] MITS researchers need to be acutely aware of inadvertent social harms that CoD results might bring to family members, particularly mothers. Similar to HIV research, MITS investigators should prioritise risk/benefit analyses of MITS participation to promote research participant welfare.[24 25] As MITS becomes more widely used, experiences of families who have consented to MITS will potentially contribute to wider community acceptability. If families are unable to share their experiences due to fear of social harms, this could negatively influence MITS uptake and normalisation of the procedure.

One concerning finding was that some participants were informed, prior to consent approach, by HCWs outside of the research team, of transportation and coffin provision through the MiM study, raising concerns of tacit coercion, although driven by in-hospital rumour rather than direct and purposeful communication from study staff. Non-study HCWs presented the study as an opportunity for families with limited resources to receive assistance—trying to help families in their time of need while going against the research protocol which outlined offering transportation and coffin after the approach for consent and regardless of the consent decision. Re-training of HCWs regarding the MiM consent protocol appears to successfully mitigate this practice. Research participation in resource limited settings is often influenced by perceived and actual benefits.[26] Transportation is a much-needed service in Malawi, especially after death because public transit will not transport a body, resulting in transport provision alone to be a sufficient explanation for research participation. This raises ethical concerns around undue inducement, while also highlighting the acute need for support and resources after losing a loved one.[27 28] However, fear of undue inducement can lead to under-compensation or failure to offer essential support for families facing significant difficulties. As such, future MITS studies should carefully consider the benefits provided to all families (regardless of participation) to minimise undue inducement and also support families in need. Researchers in general should carefully monitor how potential participants are introduced to the study team and whether benefits are presented in advance of the informed consent procedures by non-researchers, especially if compensation is only provided for those that consent.

Participants' limited ability to describe the MITS procedures 6 weeks later also raises questions about the ability to process information soon after the loss of a child and to revisit that information later. One study examining five other MITS sites in sub-Saharan Africa and South Asia reported emotional turmoil after loss of a loved one associated with limited understanding of the procedure during the consent process, requiring additional time for deliberation to consent.[29] While interviewees in this study reported understanding participation as voluntary, they focused more on the benefits and potential outcome of the procedures (improved CoD determination) rather than the procedure itself. Prior to study implementation, formative research was conducted through focus group discussions with HCWs, researchers, community leaders and parents as well as discussions between researchers to establish how to best present the procedures—including the level of detail—to families.[14] However, interview findings demonstrated that understanding was still limited, but participants generally perceived their understanding of information about the MITS procedure during informed consent processes as satisfactory. Other research studies conducted with bereaved parents in Australia and Canada have found that the majority of individuals feel comfortable making informed consent decisions soon after the loss of a child.[30 31]

This study has several limitations, including that we were only able to approach the first two-thirds of families for interview participation and the loss of interview data for one participant due to poor audio recording quality resulting in a small sample size. However, 100% of families approached did participate in postprocedure interviews and despite the small sample size data saturation was reached since no new themes emerged during iterative reviews of the data collected. The MiM study was paediatric and hospital-based, hence findings may not be generalisable to adult or community-based MITS studies or to settings outside of southern Malawi. Most importantly, we lack in-depth data from families who declined to consent to MITS, hence our detailed findings only reflect the views shared by family members who consented to the study.

Despite these limitations, this study provides an improved understanding of MITS decision-making factors, experiences with the consent process and plans for MITS results utilisation by family members. The importance of improved CoD determination was highlighted as an important benefit of MITS that supersedes some potential barriers to participation, including negative interactions and poor communication with HCWs. This study highlights the importance of monitoring MITS and other studies for unintended undue inducement while also providing adequate support for grieving families with limited financial resources. The study also raises important questions around best practices for information provision in MITS studies to balance sufficient detail for informed consent during an emotionally turbulent period when such content may be less easily absorbed and/or remembered. As MITS studies become more prevalent, the ethical issues raised in this study should continue to be explored and delineated, including across different geographic and cultural contexts.

**Author affiliations**
[1]Pediatrics, University of Washington, Seattle, Washington, USA
[2]Behaviour and Health Research Group, Abertay University, Dundee, UK
[3]Behavior and Health Research Group, Malawi-Liverpool-Wellcome Trust Clinical Research Programme, Blantyre, Malawi
[4]Global Health, University of Washington, Seattle, Washington, USA
[5]Behaviour and Health Research Group, Malawi-Liverpool-Wellcome Trust Clinical Research Programme, Blantyre, Malawi
[6]Paediatrics, University of Malawi College of Medicine, Blantyre, Malawi
[7]KEMRI Wellcome Trust Research Programme, The Childhood Acute Illness & Nutrition Network, Nairobi, Kenya
[8]Health Systems Collaborative, Nuffield Department of Medicine, University of Oxford, Oxford, UK
[9]Pediatrics, Amsterdam University Medical Centres, Amsterdam, The Netherlands
[10]Department of Pediatrics, University of Washington, Seattle, Washington, USA
[11]International Public Health, Liverpool School of Tropical Medicine, Liverpool, UK

**Contributors** ND, DD, WV, SM, MK and CH conceived the initial study design. DN, AH and DC were involved in data collection and data processing. DN and SL managed the data. SL, DN and SBM contributed to data analysis. SL, DN, ND and DD led the writing of the paper. All authors read and approved the final manuscript.

ND is the guarantor responsible for overall work and conduct of this study, access to the data and controlled the decision to publish.

**Funding** This study was funded by the Bill & Melinda Gates Foundation (https://www.gatesfoundation.org/) as a substudy, led by DD and WV, of the CHAIN Network (grant number OPP1131320). Additional funding for MK and SM provided by a Wellcome Trust and MRC Newton Fund Collaborative Award (https://mrc.ukri.org/news/browse/wellcome-trust-mrc-newton-fund-collaboration/) (grant number 200344). The funders had no role in study design, data collection and analysis, decision to publish or preparation of the manuscript.

**Competing interests** None declared.

**Patient and public involvement** Patients and/or the public were not involved in the design, or conduct, or reporting, or dissemination plans of this research.

**Patient consent for publication** Not applicable.

**Ethics approval** This study involves human participants and was approved by The Malawi National Health Sciences Research Committee (Protocol #17/09/1913) and the Oxford Tropical Research Ethics Committee (Reference 34-16) approved the study protocols. The University of Washington Institutional Review Board (STUDY00003689) exempted the study from review. Written informed consent was provided by all participants. Participants gave informed consent to participate in the study before taking part.

**Provenance and peer review** Not commissioned; externally peer reviewed.

**Data availability statement** The data that support the findings of this study are available on request from the corresponding author, ND. The data are not publicly available due to their containing information that could compromise the privacy of research participants.

**ORCID iDs**
Donna Denno http://orcid.org/0000-0002-5968-9266
Nicola Desmond http://orcid.org/0000-0002-2874-8569

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
