## [Reviewer comments · BMJ Open]

ARTICLE DETAILS

TITLE (PROVISIONAL)	Primary motivations for and experiences with pediatric minimally invasive tissue sampling (MITS) participation in Malawi: A qualitative study
AUTHORS	Lawrence, Sarah; Namusanya, Dave; Mohamed, Sumaya; Hamuza, Andrew; Huwa, Cornelius; Chasweka, Dennis; Kelley, M; Molyneux, Sassy; Voskuil, Wieger; Denno, Donna; Desmond, Nicola

VERSION 1 – REVIEW

REVIEWER	Shiyam Tikmani Aga Khan University, Community Health Sciences
REVIEW RETURNED	01-Feb-2022

GENERAL COMMENTS	I congratulate the author and team to conduct this important study. I have a few comments/suggestions The title of the study is too long - please rephrase for more clarification The objectives are not in sync with the title - please rephrase the objectives After reading, the study design seems to be a mixed-method qualitative study. Is it so? Then say it Please rephrase page 6 line 115-116 You have mentioned on page 9 lines 192, 193 that interview was conducted with the mother and father together. Do you think the father has an undue influence on the response to the mother? Why did you retake the interview with the mother separately? How this data was handled is unclear. The discussion should be focused on the implication for ethical practice in MITS research as mentioned in the title.
---

REVIEWER	Manoja Das The INCLEN Trust International
REVIEW RETURNED	06-Feb-2022

GENERAL COMMENTS	The authors present an important aspect of MITS acceptance, the factors responsible and understanding after 6 weeks in Malawi. The manuscript can be improved if the following issues are addressed. 1. Methods1.1. How was data saturation assessed?1.2. Was the interview conducted with parents together or father and/or mother separately?2. Results2.1. Did the CoD mentioned after MITS results change compared to that told to them at the time of death for some participants? If so, what was the reaction of parents for the same?3. Discussion
--

	3.1. Statement page 20 line 438-441: "..... hypothetical findings from formative MITS acceptability research in Mozambique, South Africa, and Pakistan. [6, 8, 9]. Also, similar observations have been reported from Bangladesh and India. You may add the references - Shiyam Sunder Tikmani, Sarah Saleem, Janet L Moore, Sayyeda Reza, Guruprasad Gowder, Sangappa Dhaded, S Yogesh Kumar, Shivaprasad S Goudar, Vardendra Kulkarni, Sunil Kumar, Anna Acetuino, Lindsay Parlberg, Elizabeth M McClure, Robert L Goldenberg, Factors Associated With Parental Acceptance of Minimally Invasive Tissue Sampling to Identify the Causes of Stillbirth and Neonatal Death, Clinical Infectious Diseases, Volume 73, Issue Supplement_5, 15 December 2021, Pages S422–S429, https://doi.org/10.1093/cid/ciab829 - Das MK, Arora NK, Kaur G, Malik P, Kumari M, Joshi S, Rasaily R, Chellani H, Gaikwad H, Debata P, Meena KR. Perceptions of family, community and religious leaders and acceptability for minimal invasive tissue sampling to identify the cause of death in under-five deaths and stillbirths in North India: a qualitative study. Reprod Health. 2021 Aug 4;18(1):168. doi: 10.1186/s12978-021-01218-4. PMID: 34348749; PMCID: PMC8336381. Das MK, Arora NK, Debata P, Chellani H, Rasaily R, Gaikwad H, Meena KR, Kaur G, Malik P, Joshi S, Kumari M. Why parents agree or disagree for minimally invasive tissue sampling (MITS) to identify causes of death in under-five children and stillbirth in North India: a qualitative study. BMC Pediatr. 2021 Nov 17;21(1):513. doi: 10.1186/s12887-021-02993-6. PMID: 34784903; PMCID: PMC8597286. 3.2. For the statement page 20 line 444-446: "This is in contrast to Reference from the Indian study finding regarding the influence of HCW relationship and trust on them may be added. - Das MK, Arora NK, Kaur G, Malik P, Kumari M, Joshi S, Rasaily R, Chellani H, Gaikwad H, Debata P, Meena KR. Perceptions of family, community and religious leaders and acceptability for minimal invasive tissue sampling to identify the cause of death in under-five deaths and stillbirths in North India: a qualitative study. Reprod Health. 2021 Aug 4;18(1):168. doi: 10.1186/s12978-021-01218-4. PMID: 34348749; PMCID: PMC8336381. 3.3. Limitations Should add small sample size as a limitation.
--	--

VERSION 1 – AUTHOR RESPONSE

Reviewer: 1

Dr. Shiyam Tikmani, Aga Khan University

Comments to the Author:

3. I congratulate the author and team to conduct this important study. I have a few comments/suggestions. The title of the study is too long - please rephrase for more clarification.

Thank you, we have revised the title to “Primary motivations for and experiences with pediatric minimally invasive tissue sampling (MITS) participation in Malawi: A qualitative study.”

4. The objectives are not in sync with the title - please rephrase the objectives

Thank you for this comment. We have instead revised the title to align better with the existing study objectives.

5. After reading, the study design seems to be a mixed-method qualitative study. Is it so? Then say it
Please rephrase page 6 line 115-116

Thank you, the study is not a mixed-method qualitative study.

6. You have mentioned on page 9 lines 192, 193 that interview was conducted with the mother and father together. Do you think the father has an undue influence on the response to the mother? Why did you retake the interview with the mother separately? How this data was handled is unclear.

Thank you for raising this question. We have added text on lines 132-134 on page 6 that notes, "Interview participants were determined by the family and family members interviewed together as is natural in the cultural context (see Table 1)." We have revised the original text on page 9 (now page 8 lines 196-197) to clarify that mothers were not interviewed alone after a joint interview with both parents. This was meant to reflect the second most common interview scenario was a mother attending alone.

We do not believe the fathers had undue influence over the mother's during the interviews, although they were often the more dominant speaker mother's shared their own unique perspectives.

7. The discussion should be focused on the implication for ethical practice in MITS research as mentioned in the title.

As we did not discuss this extensively, we have removed this from the title.

Reviewer: 2

Dr. Manoja Das, The INCLIN Trust International

Comments to the Author:

The authors present an important aspect of MITS acceptance, the factors responsible and understanding after 6 weeks in Malawi. The manuscript can be improved if the following issues are addressed.

1. Methods

1.1. How was data saturation assessed?

Thank you, we now note on page 23, lines 512-513 that saturation was reached with our small sample size as no new themes emerged during iterative reviews of the data collected.

1.2. Was the interview conducted with parents together or father and/or mother separately?

Thank you for raising this question. We interviewed mothers and fathers together if both were available, per family preference and cultural norms.

We have added text to clarify this on lines 132-134 on page 6 that notes, "Interview participants were determined by the family and family members interviewed together as is natural in the cultural context (see Table 1)" and text on page 8, "Interviews included one to three family members as preferred by participants, most often the mother and father together if both were available (Table 1)."

2. Results

2.1. Did the CoD mentioned after MITS results change compared to that told to them at the time of death for some participants? If so, what was the reaction of parents for the same?

Thank you for this question. Participants often reported they were not told a specific CoD at the time of death, but could sometimes identify diagnoses available before the child's death. No participant compared and contrasted CoD before death or after MITS.

3. Discussion

3.1. Statement page 20 line 438-441: "..... hypothetical findings from formative MITS acceptability research in Mozambique, South Africa, and Pakistan. [6, 8, 9].

Also, similar observations have been reported from Bangladesh and India.

You may add the references

- Shiyam Sunder Tikmani, Sarah Saleem, Janet L Moore, Sayyeda Reza, Guruprasad Gowder, Sangappa Dhaded, S Yogesh Kumar, Shivaprasad S Goudar, Vardendra Kulkarni, Sunil Kumar, Anna Acetuino, Lindsay Parlberg, Elizabeth M McClure, Robert L Goldenberg, Factors Associated With Parental Acceptance of Minimally Invasive Tissue Sampling to Identify the Causes of Stillbirth and Neonatal Death, Clinical Infectious Diseases, Volume 73, Issue Supplement_5, 15 December 2021, Pages S422–S429, <https://doi.org/10.1093/cid/ciab829>

- Das MK, Arora NK, Kaur G, Malik P, Kumari M, Joshi S, Rasaily R, Chellani H, Gaikwad H, Debata P, Meena KR. Perceptions of family, community and religious leaders and acceptability for minimal invasive tissue sampling to identify the cause of death in under-five deaths and stillbirths in North India: a qualitative study. *Reprod Health*. 2021 Aug 4;18(1):168. doi: 10.1186/s12978-021-01218-4. PMID: 34348749; PMCID: PMC8336381.

Das MK, Arora NK, Debata P, Chellani H, Rasaily R, Gaikwad H, Meena KR, Kaur G, Malik P, Joshi S, Kumari M. Why parents agree or disagree for minimally invasive tissue sampling (MITS) to identify causes of death in under-five children and stillbirth in North India: a qualitative study. *BMC Pediatr*. 2021 Nov 17;21(1):513. doi: 10.1186/s12887-021-02993-6. PMID: 34784903; PMCID: PMC8597286.

Thank you, we have added these references on page 20, line 444.

3.2. For the statement page 20 line 444-446: "This is in contrast to

Reference from the Indian study finding regarding the influence of HCW relationship and trust on them may be added.

- Das MK, Arora NK, Kaur G, Malik P, Kumari M, Joshi S, Rasaily R, Chellani H, Gaikwad H, Debata P, Meena KR. Perceptions of family, community and religious leaders and acceptability for minimal invasive tissue sampling to identify the cause of death in under-five deaths and stillbirths in North India: a qualitative study. *Reprod Health*. 2021 Aug 4;18(1):168. doi: 10.1186/s12978-021-01218-4. PMID: 34348749; PMCID: PMC8336381.

Thank you, we have added this reference on page 20, lines 444 and 449.

3.3. Limitations

Should add small sample size as a limitation.

Thank you, we have added this on page 23, line 513.

VERSION 2 – REVIEW

REVIEWER	Manoja Das The INCLEN Trust International
REVIEW RETURNED	10-May-2022
GENERAL COMMENTS	The authors have addressed the comments.